# Three-dimensional domain identification in a single hexagonal manganite nanocrystal

Ahmed H. Mokhtar [1] ✉, David Serban [1], Daniel G. Porter [2], Frank Lichtenberg[3], Stephen P. Collins[2], Alessandro Bombardi [2], Nicola A. Spaldin [3] & Marcus C. Newton [1] ✉

The three-dimensional domain structure of ferroelectric materials significantly influences their properties. The ferroelectric domain structure of improper multiferroics, such as YMnO₃, is driven by a non-ferroelectric order parameter, leading to unique hexagonal vortex patterns and topologically protected domain walls. Characterizing the three-dimensional structure of these domains and domain walls has been elusive, however, due to a lack of suitable imaging techniques. Here, we present a multi-peak Bragg coherent x-ray diffraction imaging determination of the domain structure in single YMnO₃ nanocrystals. We resolve two ferroelectric domains separated by a domain wall and confirm that the primary atomic displacements occur along the crystallographic c-axis. Correlation with atomistic simulations confirms the Mexican hat symmetry model of domain formation, identifying two domains with opposite ferroelectric polarization and adjacent trimerization, manifesting in a clockwise arrangement around the hat's brim.

Ferroelectric materials have a spontaneous electric polarization that is switchable by the application of an electric field, making them useful in technological applications such as storage and transduction[1]. The switching proceeds via the motion of the domain walls separating domains of different polarization orientations, and a detailed description of the three-dimensional structure of the domains and domain walls is invaluable for optimizing the switching process, as well as for engineering domain walls as functional entities in their own right[2]. The domain structure in the class of improper multiferroic hexagonal manganites, RMnO₃ (R = Y, Sc or rare earth), is of particular interest since the improper nature of the ferroelectricity[3,4] combines with the hexagonal symmetry to yield an unusual six-fold pattern of alternating polarization separated by topologically protected domain walls around string-like vortex cores[5]. These nanometer-sized topological defects, which can be controlled using electric fields, have been shown to exhibit electrical conductivity and magnetic properties, suggesting new pathways to novel devices such as mechanical sensors, transducers, and memories[6,7]. The domain structure in the hexagonal manganites is formed on cooling through the symmetry-lowering structural phase transition (SPT) from the high symmetry paraelectric $P6_3/mmc$ phase to the ferroelectric $P6_3cm$ phase at high temperature; in the prototypical yttrium manganite, YMnO₃ (YMO), this occurs at $T_c \sim 1250$ K[8]. In the high symmetry structure, triangular planes of $Y^{3+}$ ions separate planes of corner-shared MnO₅ trigonal bipyramids perpendicular to the crystallographic c axis, Fig. 1a. The transition is driven by a unit-cell-tripling zone-boundary $K_3$ mode of the high-symmetry unit cell, which consists of a trimerizing tilting of the MnO₅ polyhedra accompanied by a buckling of the $Y^{3+}$ ions along the c direction, Fig. 1b. Coupled with the $K_3$ mode is a polar $\Gamma_2^-$ mode consisting of a uniform displacement of all Y ions in the same direction that causes an electric polarization along the c axis[4,9]. The resulting energy surface resembles the iconic $\phi^4$ Mexican hat potential, Fig. 1d, but with six additional local minima in its brim at polyhedral tilting angles $\phi = \frac{n\pi}{3}$, corresponding to the six low-symmetry structural domains, often designated α+, β−, γ+, α−, β+ and γ−, where α, β and γ indicate three equivalent choices of origin for the tilting and +, − indicate the polarization direction, Fig. 1c[10].

¹School of Physics and Astronomy, University of Southampton. University Road, Southampton SO17 1BJ, UK. ²Beamline I16, Diamond Light Source. Harwell Science and Innovation Campus, Didcot, Oxfordshire OX11 0DE, UK. ³Department of Materials, ETH Zurich. Ramistrasse 101, 8092 Zurich, Switzerland. ✉e-mail: ahmm1g15@soton.ac.uk; m.c.newton@soton.ac.uk

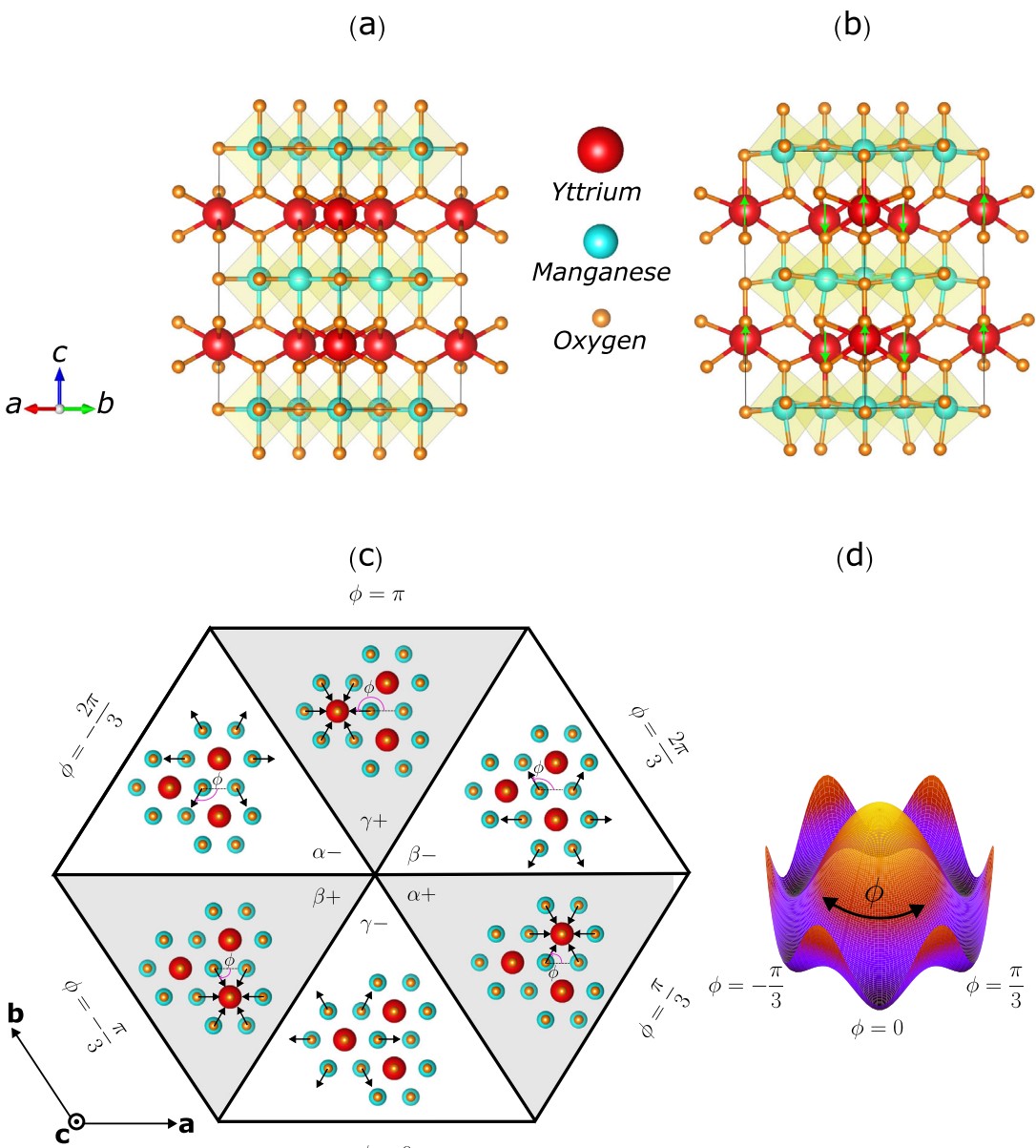

**Fig. 1 | Material structure characteristics. a, b** Side-view of the paraelectric and ferroelectric unit cells, respectively. Yttrium (Y) ions are red, Oxygen (O) ions are orange, and Manganese (Mn) ions are cyan. The green arrows on the Y ions indicate the direction of displacements from the centrosymmetric phase. **c** Illustration of the six structural domains in YMnO$_3$ viewed down the c-axis, with the arrows indicating the displacements of the apical Oxygen ions corresponding to the tilting of the MnO$_5$ polyhedra. **d** Potential energy surface of the YMnO$_3$ structural phase transition in YMnO$_3$. The centrosymmetric paraelectric state's energy is at the hat's peak. The six equivalent ferroelectric structures of (**c**) correspond to the six minima in the potential, with trimerisation angle $\phi$ going around the hat. Crystal structure visualizations presented in **a–c** are prepared using the VESTA software[44].

Analogies have therefore been drawn between the domain formation process in the hexagonal manganites and other fundamental physical processes that are described by such a Mexican hat potential, with the hexagonal manganites proposed as a laboratory-based simulator for the formation of cosmic strings in the early universe[11,12], as well as for aspects of Higgs-Goldstone physics[13,14]. These fundamental and technological aspects clearly motivate a full three-dimensional mapping of the domain structure in the hexagonal manganites. Currently, however, this is lacking, with information about the sample interior inferred from microscopy[15] or scanning probe measurements[16] of the intersection of the domains with the surface, or from computer simulations.

Bragg coherent X-ray diffraction imaging (BCDI) is a particularly powerful tool for this purpose as it can reveal the ferroelectric domain structure in three dimensions as well as emerging crystal defects[17–19]. BCDI is performed by illuminating a sample with a spatially coherent X-ray source so that the coherence length exceeds the dimensions of the crystal[20,21]. Scattered light from the entire volume of the crystal interferes in the far field, producing a three-dimensional k-space diffraction pattern[22]. The experiment collects the 2D diffraction pattern of a selected reflection onto a detector while the 3rd dimension is obtained by rocking the sample in increments and collecting the diffraction pattern at each step.

The Fourier space density and the real space electron density are related to each other by Fourier transforms, however, since the experiment only measures the intensity of the diffraction pattern, the phase information is lost as a result[23,24]. Iterative phase retrieval methods like Hybrid Input-Output (HIO) algorithm are then used to

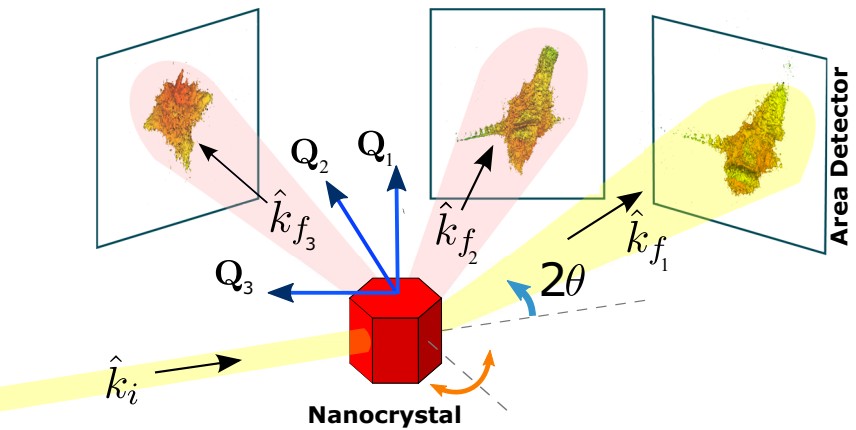

**Fig. 2 | Schematic layout of the experiment.** An illustration of the experimental geometry in a multi-Bragg Coherent X-Ray Diffraction Imaging (BCDI) experiment. The three detector positions show real data taken in this experiment, from left to right: the (212), (111) and (110) diffraction data.

recover the complex three-dimensional electron density and phase information[25-27]. The algorithm alternates between applying Fourier and real-space constraints on the electron density until it converges to a stable solution that satisfies the diffraction constraints[28,29].

The reconstructed real space phase information, $\phi(\mathbf{r})$, reveals a projection of the atomic displacement, $\mathbf{u}(\mathbf{r})$, at a point $\mathbf{r}$ in the crystal from equilibrium along the direction of the chosen $\mathbf{Q}$-vector in the experiment, according to $\phi(\mathbf{r}) = \mathbf{Q} \cdot \mathbf{u}(\mathbf{r})$. Hence, the real space phase enables the retrieval of atomic displacement parallel to the Bragg peak reciprocal lattice vector. The derivative of this displacement along the same vector provides the normal strain field, which serves as a proxy for identifying structural changes within the material. However, for a single Bragg peak, only a single projection of the displacement field is obtained. Consequently, components of the displacement field perpendicular to the scattering vector $\mathbf{Q}$ will not be observed in the BCDI measurement. To recover the full displacement field and strain tensor, diffraction patterns from multiple non-planar $\mathbf{Q}$-vector directions are required[30].

In this paper, we image the domain structure of a YMO single nanocrystal by recovering the spatially resolved full strain tensor field using 3D BCDI.

## Results

### Multi-peak BCDI experiment
The experimental details of the preparation of the melt-grown crystalline YMO are reported in ref. 31. Nanocrystals were then synthesized using a mixture of top-down and bottom-up approaches (see Methods section).

The BCDI experiment took place in air on beamline I16 at the Diamond Light Source synchrotron facility using x-rays of 9 keV energy in the Bragg geometry on a 6-axis kappa diffractometer. The beam size was focused down to $200 \times 30\,\mu m$ with front slits set to $20 \times 20\,\mu m$. Multiple Bragg reflections from a single nanocrystal were located as described in the Methods section. Rocking curve measurements were performed on each reflection to obtain the three-dimensional diffraction patterns. In total, 5 specular reflections were identified, namely the (111), (110), (11$\bar{1}$), (212), and (300) reflections.

The precision in our diffraction measurements is underscored by the close match between the observed $2\theta$ values and their calculated counterparts for the studied reflections using the reported lattice parameters[32]. This match is quantitatively expressed by the quantity $\sum \frac{|2\theta_{obs} - 2\theta_{calc}|}{2\theta_{calc}} < 10^{-3}$ indicating a high level of accuracy. This correspondence not only confirms the accuracy of our measurements but also validates the correct phase and stoichiometry of the material.

We also investigated the fringe frequency in our diffraction patterns to confirm that all patterns originated from the same crystal.

Given the random size and morphology of nanocrystals on the substrate, we needed to ensure the fringe frequencies across the diffraction patterns were consistent. To this end, we performed a line scan in the same direction for all five diffraction patterns and computed the Fourier transform of these scans to determine the frequency. The results were similar, with frequencies centered around 250 nm, Supplementary Fig. 2. Thereby confirming that the patterns originated from the same crystal as the likelihood of two crystals having identical orientation matrices and sizes in a randomly distributed sample is exceedingly low. The calculated fringe frequency matches one of the reconstructed crystal's dimensions, as expected, providing further validation that the diffraction patterns originate from the same crystal.

### BCDI concurrent phase retrieval
Conventional BCDI is typically performed on a single reflection. However, advancements in the field have facilitated the development of Multi-BCDI. This innovative approach allows for the concurrent analysis of multiple Bragg peaks from a single nanoparticle, thereby enabling the effective reconstruction of vector-valued lattice distortion fields within nanoscale crystals[33-36].

Reconstructions of the full displacement field were performed concurrently[33] using *The Interactive Phase Retrieval Suite*[37] by a combination of Fienup's HIO Mask[29] and Error Reduction[38] algorithms, with the support created using a manual version of the shrink wrap method. It accounts for geometric factors arising in BCDI measurements and allows for a globally constrained single image reconstruction to multiple Bragg peak measurements.

Additionally, our method involved an additional real space constraint based on the mean of the different amplitudes concurrently reconstructed from multiple Bragg reflections, since these reflections originated from the same crystal. This step was instrumental in accelerating the convergence of our phase retrieval process, enhancing the accuracy and efficiency of our reconstruction of the 3D strain field. During the reconstruction, it was realized that the diffraction patterns (11$\bar{1}$) and (300) reflections had very poor signal-to-noise ratios, hence, they were omitted from the reconstruction.

The other three patterns, displayed in Fig. 2, contained sufficient information for the full reconstruction of the full strain tensor. The full reconstruction proceeded for 10,000 iterations consisting of periodic cycles of HIO Mask and ER Mask algorithms in a 10:1 ratio. Importantly, the requirement for extracting a full 3D strain tensor is not the orthogonality of these peaks, but their non-coplanarity. The non-coplanar arrangement of the three Bragg peaks in our experimental setup is a key factor that enables the effective extraction of the full 3D strain tensor.

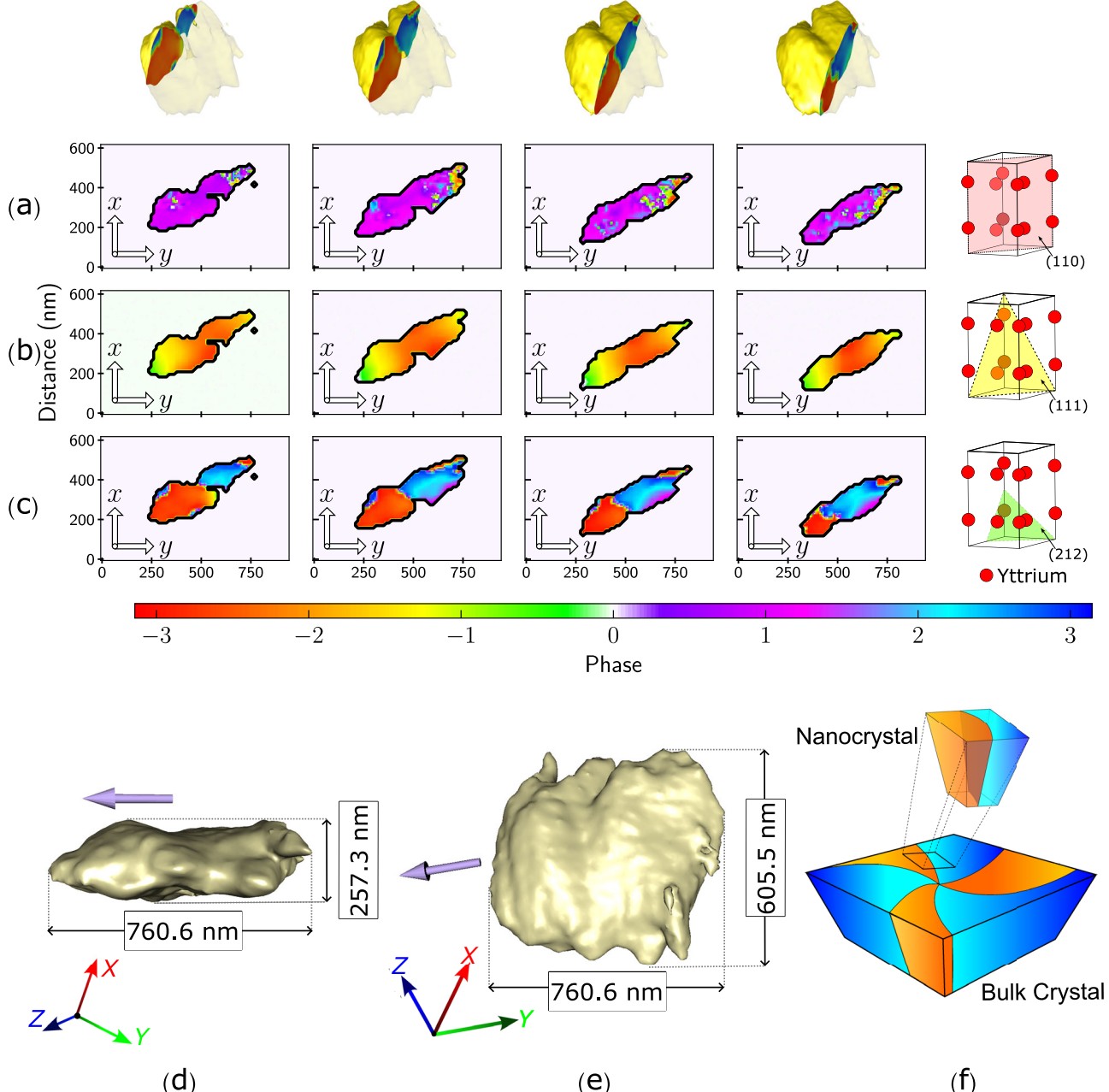

**Fig. 3 | The reconstructed amplitude and phase.** 2D slices of the three reconstructed phase maps: **a** the (111), **b** (110) and **c** (212) reflections. The slices of the different phase maps are taken at the same angle (perpendicular to the z-axis) and the same locations in the crystal. The 3D rendering of the crystal on the top row indicates the position at which the slices are taken. The last column illustrates the different planes for each reflection and displays the Yttrium ions' positions in the unit cell. **d**, **e** 3D renderings of the reconstructed crystal were taken from two different viewpoints displaying 70% of the amplitude. The (212) **Q**-vector displayed in purple. **f** An illustration of how the two observed domains form part of the six intersecting domains in the bulk crystal.

Additionally, we observe that separate domains are encoded within the same diffraction pattern without separation in Fourier space which arises due to the uniformity of the c/a ratio across different domains. The distinguishing feature among these domains lies in the varied origins of rotation for the $MnO_5$ polyhedra. This aspect notably contrasts with other commonly studied ferroelectrics, where an anisotropic c/a ratio results in distinct pairs of Bragg reflections for each domain in reciprocal space[39]. Our atomistic simulations, discussed in subsequent sections of the paper, provide further corroboration of this characteristic in $YMnO_3$.

## BCDI characterization of ferroelectric domains

The 3D reconstruction of the electron density, Fig. 3d, e reveal the dimensions of the crystal to be $761 \times 605 \times 257 \, nm^3$, with a 13 nm resolution which includes the 5–10 nm layer added by PLD. The orientation of the c-axis was determined using the known directions of the **Q**-vectors and the geometry of the unit cell. For simplicity in data interpretation, we then rotated the crystal such that its c-axis aligned with the z-axis. A 3D rendering of the reconstructed displacement field is presented in Supplementary Movie 1.

Sliced planes were then taken at different positions along the crystal, perpendicular to the z-axis, to obtain phase maps for each of the reflections; Fig. 3a–c shows slices of the x-y plane. The full phase map is presented in Supplementary Figs. 3 and 4. The added layer by PLD does not manifest itself in the phase information nor does it influence the domain structure observed or the overall phase of the material, hence, it can be neglected.

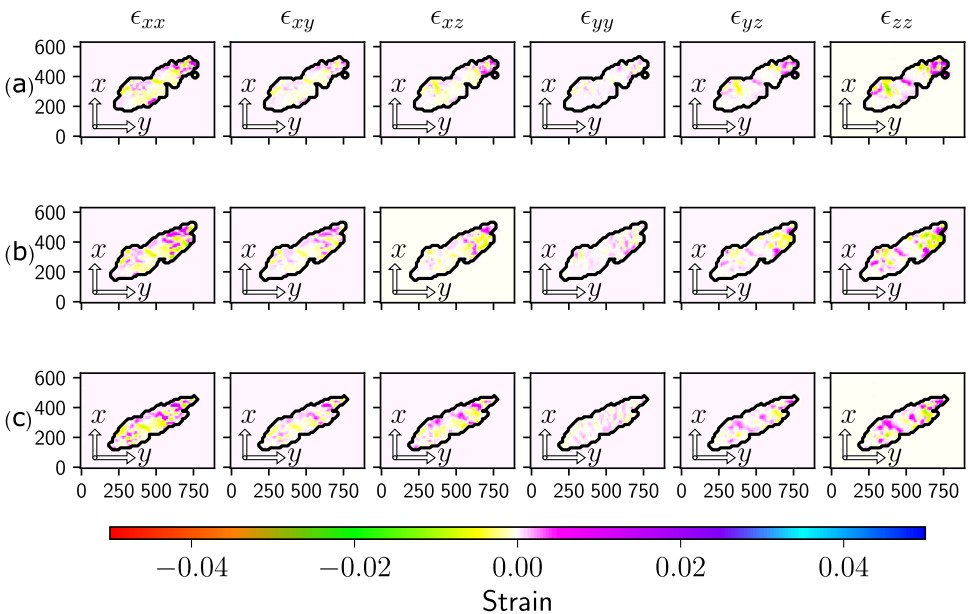

**Fig. 4 | The reconstructed full strain tensor.** The reconstructed strain tensor of the crystal, computed as described in the text. The six columns represent each of the six independent components of the strain tensor. Each row is a cross-sectional plane in the crystal taken at the **a** first, **b** second and **c** third locations of the four cross-sectional planes in Fig. 3.

The phase map along (212) reveals the existence of two domains with high phase values and hence large displacement along that **Q**-vector direction. Domain 1 exhibits a mean phase value of $2.25 \pm 0.006$ rad, while Domain 2 has a mean phase value of $-2.54 \pm 0.004$ rad. The near symmetry in the phase values and their low standard error indicate similar magnitudes of lattice distortions in opposite directions. This is characteristic of a domain boundary rather than dislocation-induced lattice distortion.

In contrast, the phase information of dislocations would manifest as localized disruptions in the crystal lattice and a gradual phase change away from the dislocation site due to lattice relaxation[40,41]. The phase changes observed here are step-like, with each domain exhibiting a low standard error which is indicative of a small phase gradient.

Additionally, in our work, the phase contrast is predominantly visible in the (212) direction, and notably absent in the (110) and (111) phase maps. This selective visibility supports the interpretation that (212) contrasts are indicative of domain boundaries rather than dislocations, as a dislocation would likely be visible in the other phase maps. Additionally, the selective visibility is consistent with our atomistic simulations discussed later in the paper.

Furthermore, the crystal shows no obvious signs of Oxygen depletion at the surface when scaled down to the nanoscale. The absence of Oxygen diffusion is consistent with the bulk material[31]. Typically, Oxygen depletion would manifest as a distinct phase at the crystal's surface, but this is not observed in our results, further supporting the integrity of our material preparation.

The locations and widths of these (212) boundaries differ from the phase contrast observed in the (110) reflection reflecting their distinct origins. The structure along the (110) axis reveals a boundary made up of small phase values implying that there is a small displacement along that direction in this region. The neighboring region exhibits slightly elevated phase values, indicating a larger displacement. The (110) phase map is reflective of displacements in the crystal's ab-plane likely caused by some small crystal imperfections or secondary strain effects.

In the (111) phase map, no distinction between the domains was observed in the (111) phase map despite the **Q**-vector having a component along the c-axis. This lack of distinction can be attributed to the symmetry of Yttrium ion displacement around (111) **Q**-vector direction, which effectively masks the differentiation between domains in the (111) phase map. This observation is corroborated by the atomistic simulations discussed later in the paper, which further illustrate the symmetrical nature of these displacements when projected onto that direction.

To further confirm the origin of these domains we computed the strain tensor field in the crystal. The strain tensor was computed from the reconstructed displacement field according to Eq. (1),

$$\epsilon_{ij} = \frac{1}{2}\left(\frac{\partial \mathbf{u}_i}{\partial x_j} + \frac{\partial \mathbf{u}_j}{\partial x_i}\right) \tag{1}$$

where $\epsilon$ represents the strain tensor, and subscripts $i, j$ indicate the different components of the tensor. Figure 4 displays slices of the 6 independent components of the strain tensor taken at three of the four planes presented in Fig. 3. The full strain tensor field is presented in Supplementary Figs. 5 and 6.

The domain wall is most evident in the $\epsilon_{zz}$ and $\epsilon_{yz}$ components as illustrated in Fig. 4. It is marked by distinct purple strings and is notably absent in the $\epsilon_{xx}$ and $\epsilon_{yy}$ components. The strings align with the domain wall identified in the (212) phase map, highlighting the displacement direction between different domains. The c-axis vector aligns with the z-axis which suggests that the observed displacement between domains is primarily due to Yttrium ion movement along the c-axis. In the strain tensor, dislocations typically exhibit circular wraps and a profile of gradual increase or decrease. Therefore, the observed profile further supports the existence of ferroelectric domains.

Piezoforce microscopy (PFM) images of the bulk crystal[31] reveal the density of vortex cores at the surface to be $0.3$ cores $\cdot \mu m^{-2}$. Given this vortex density and the small size of the crystal, observation of a complete vortex or multiple vortices is improbable. Therefore, we are observing a small portion of the full structure, a section that contains two domains separated by a domain wall as illustrated in Fig. 3f.

Our investigations reveal that the vortex structure within the crystal is independent of the crystal's size as the topological structures remain unaffected by variations in the crystal's dimensions. Notably, the stability and consistency of the vortex size, irrespective of the

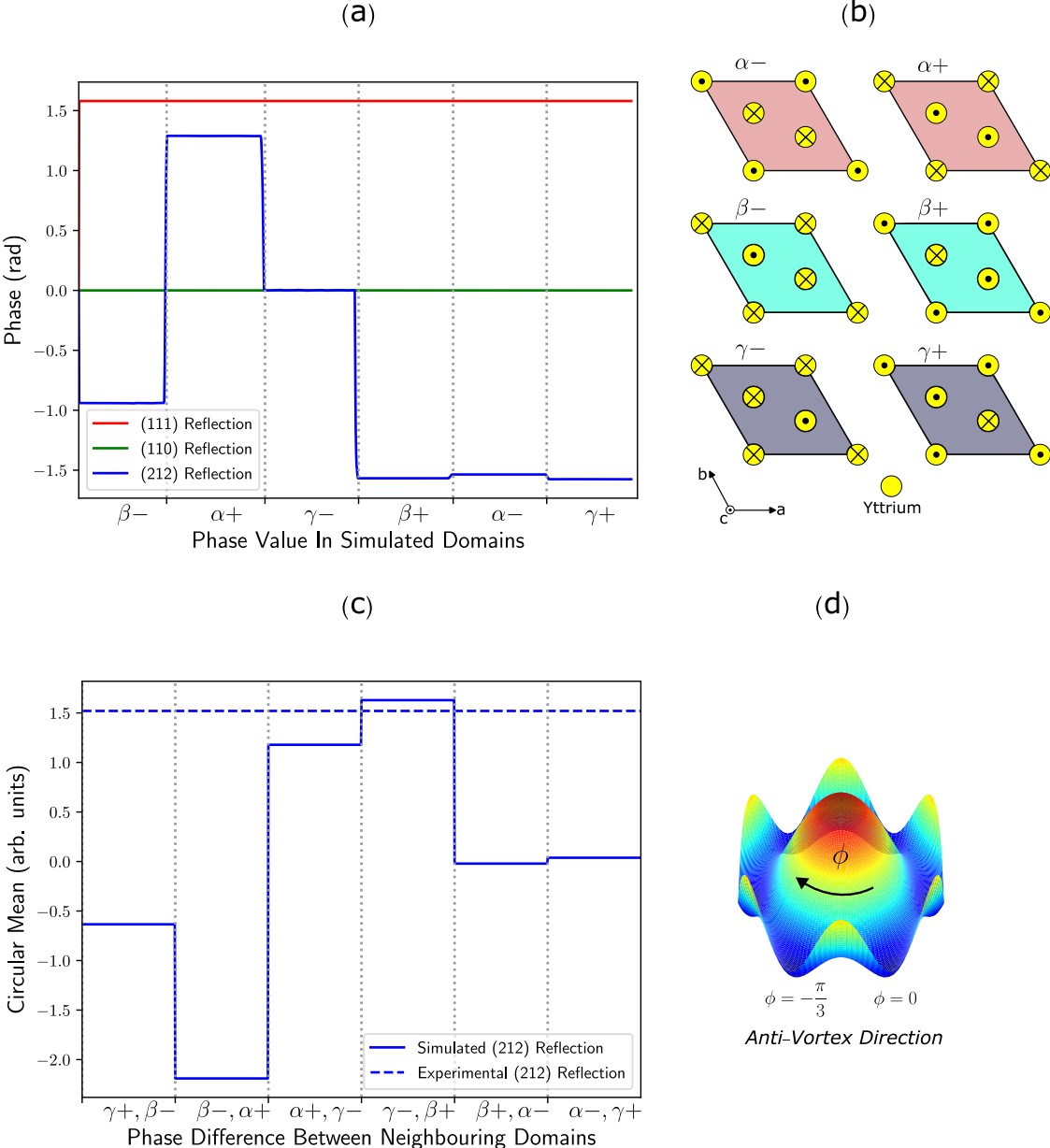

**Fig. 5 | Atomistic simulation of the domain structures. a** A plot of the scan through the simulated crystal showing the phase value of the different domains. **c** A plot of the calculated circular variance between the phase of neighboring domains for the simulated (colored solid lines) and experimental (colored dotted line). The black vertical dotted lines separate the different regions, as indicated in the figure. A match between the simulated and experimental values for the (212) reflection at the β+ and γ− region conclude the type of domains observed experimentally. **b** Illustrations of the ions' movements in the ferroelectric unit cell relative to their positions in the paraelectric phase for the Yttrium ions. **d** The clockwise arrow indicates an anti-vortex structure, as is the case for the observed crystal.

crystal size, are anticipated based on the topologically protected nature of these structures, suggesting that their characteristics are fundamentally governed by topological constraints rather than geometrical dimensions of the crystal.

**Atomistic simulations of BCDI for domain type identification**
Our focus is primarily on the order parameter of the different domains exhibited in the (212) phase map and the direction of its change, which we have determined to be along the c-axis. Using this information, the domain types in the (212) phase can be determined using an atomistic simulation of BCDI 3D diffraction patterns with lattice distortions as described in the Methods section[42].

We consider the Yttrium and planar Oxygen ions' displacements along the c-axis in the 6 domains relative to their positions in the centrosymmetric paraelectric phase. This choice is supported by the strain field information that indicates the major displacement between the domain occurs along that direction. An arbitrary choice of origin is selected and fixed for all the simulated domains while the ions are displaced differently in each of the domains, leading to an overall change in the momentum transfer vector along certain **Q**-vector directions. The displacement values of the Yttrium ions in the ferroelectric unit cell have been previously detailed[32], with the Yttrium ions' movements for the choice of origin illustrated in Fig. 5b and the planar Oxygen ions are made to displace in the opposite directions; as described in earlier studies[3,43].

The 3D diffraction pattern was simulated for the three experimental reflections and the real space phase information was obtained. Fig. 5a displays the phase information in each of the domains for each

reflection. The distinction between Freidal pairs of the **Q**-vectors is discussed in Supplementary Note 5. The (212) reflection exhibited distinguished phase information between different domains, with variations in the phase difference between the neighboring domains. This information can be used to compare the difference between the neighboring domains in the simulated crystal with the difference between the two observed experimental domains. To account for the periodic nature of the phase, the differences between domains were characterized by computing the circular mean (Eq. (2)).

$$\text{Circular Mean} = \arctan\left(\frac{\sin(\phi_i - \phi_j)}{\cos(\phi_i - \phi_j)}\right). \quad (2)$$

Within each domain in the simulated data, the phase value is constant, however, in the experimental results, a mean value was computed for each domain. Fig. 5c displays the result from these computations clearly showing a close match of the experimental and simulated circular mean at the $\gamma-$, $\beta+$ and $\alpha+$, $\gamma-$, regions with a $\chi^2$ of 0.007 and 0.071, respectively. This signifies that the $\gamma-$, $\beta+$ forms a better fit with a $\chi^2$ value that is an order of magnitude lower. It can then be determined that the two observed domains in the (212) phase map are $\beta+$ and $\gamma-$ for the positive and negative phase regions, respectively, as they form better alignment with the theoretical model.

The circular mean is a non-commutative quantity, hence, its polarity has some significance in determining the vortex direction. The circular mean values in Fig. 5c for the simulated results were computed with an anti-vortex order, hence, a matched polarity with the experimental results is indicative that the two domains observed are part of an anti-vortex structure in this case, i.e. a clockwise movement around the brim of the Mexican hat potential as illustrated in Fig. 5d.

As expected, the simulated phase in the (110) reflection was zero throughout the whole crystal as only the displacement along the crystal's c-axis was considered. The simulated results of the (111) reflection demonstrated a constant non-zero phase in the whole crystal, i.e. the domains are indistinguishable from each other along that direction. It can be said that the projection of the Yttrium ions' displacements along that **Q**-vector direction poses symmetry that detects the displacements but does not distinguish them from each other. This result is consistent with the experimental result for the (111) phase map as we observe, although not strictly uniform, a non-zero phase in addition to the domains being indistinguishable from each other. The absence of the domain primary order parameter within the experimental and the simulated results highlights the aforementioned symmetry and the reliability of our method.

## Discussion

In conclusion, we report a multi-Bragg CDI experiment on a single YMO nanocrystal and demonstrate the success of the concurrent phase retrieval algorithm. In addition, we identified a method of determining ferroelectric domain types from the reconstructed phase information. Phase maps for each of the three reflections along with the strain tensor field detail the behavior of the material.

The (212) phase map revealed two separate ferroelectric domains forming in the crystal. The two domains were identified as $\gamma-$ and $\beta+$ for our choice of origin by comparison with simulated results of the domains' structure for that reflection. We were also able to identify the two domains to be a part of an anti-vortex structure. Small displacements were observed in the (110) phase map which was attributed to crystal defects as there are no displacements of the Yttrium ions along that direction. No obvious surface defects such as Oxygen depletion or a structural change due to a depolarization field were observed in any of the phase maps.

The ability to image and identify ferroelectric domains in a single YMO nanocrystal can facilitate the design and characterization of devices formed from YMO where domains play a pivotal role. Our finds are therefore of considerable utility for the development of next-generation technologies based on multiferroic hexagonal manganite materials.

## Methods

### Synthesis of YMO nanocrystals

Nanocrystals of YMO were synthesized as follows. By means of mechanical grinding and ultra-sonication of a small piece of the melt-grown sample, a low-density nano-powder solution was formed and deposited on a bare Si (100) substrate followed by dispersion using a spin-coater to obtain an even distribution over the surface. The nano-powder acted as seeds for a pulsed laser deposition (PLD) operation aiming to correct surface imperfections caused by the grinding process and sufficient adhesion of the nanocrystal to the substrate. PLD took place in an ultra-high vacuum condition in the presence of Argon; an excimer laser was used to ablate the melt-grown YMO sample forming a plasma. This is then directed on the substrate where the material condenses at the seed sites in a stoichiometric manner. The PLD process was run for only 16 cycles at 5Hz and 600 °C such that only 5–10 nm of material is deposited. The sample was then gently annealed at 600 °C for 6 hours to relax any strain present.

### Locating multiple Bragg reflections

The detector employed in our experiments was the Quad Merlin detector, featuring $515 \times 515$ pixels, with each pixel measuring $55\,\mu m \times 55\,\mu m$. The sample-to-detector distance during measurements was 1.31 meters. The full beam x-ray flux at the sample is $3 \times 10^{13}$ photons $s^{-1}$ with a beam size of $200\,\mu m(h) \times 30\,\mu m(v)$, for these measurements we employed slits before the sample with size $20\,\mu m \times 20\,\mu m$, reducing the beam size and used Al foils to attenuate the incident flux by 70% to mitigate beam damage to the sample.

Using surface area scans, a single nanocrystal was identified at the (111) specular reflection and was positioned at the eucentric point. Using this as the primary reflection, other reflections were found by measuring the cone about the primary reflection using the inter-planar angle. Reflections were identified as being from the same grain if the inter-planar angle agreed and the peaks had the same profile. Two reflections are enough to define an orientation matrix, which was then used to find further reflections.

Prior to commencing the concurrent reconstruction of the diffraction patterns, corrections to the diffraction patterns and determination of the directions of the **Q**-vector is required. This process is detailed in Supplementary Note 1 and 2.

### Simulation of BCDI diffraction patterns

The simulation considers the displacements of individual atoms from their equilibrium position and computes the scattering amplitude according to Eq. (3),[42].

$$
\begin{aligned}
A(\mathbf{q}) &= \frac{r_0\sqrt{\phi_0}}{R} \sum_{\mathbf{R}_n} \sum_{\mathbf{r}_m} \mathcal{F}_m(\mathbf{q}) e^{-i\mathbf{q}\cdot(\mathbf{R}_n^m + \mathbf{u}_m(\mathbf{R}_n))} \\
&\approx \frac{r_0\sqrt{\phi_0}}{R} \prod_i d_i \sum_{\mathbf{R}_{nd}} \sum_{\mathbf{r}_m} \mathcal{F}_m(\mathbf{q}) e^{-i\mathbf{q}\cdot(\mathbf{R}_{nd}^m + \tilde{\mathbf{u}}_m(\mathbf{R}_{nd}))}
\end{aligned} \quad (3)
$$

where $r_0$ is the Thompson scattering factor, $\mathcal{F}_m(\mathbf{q})$ is the atomic form factor, $\mathbf{u}_m$ is the displacement of atom $m$ from equilibrium, $\mathbf{R}_n$ are unit cell vectors and $\mathbf{r}_m$ are fractional coordinates of atom $m$ in the unit cell. The computationally expensive problem of summing over all atoms is simplified by considering the summation over groups of adjacent unit cells as shown in the second line of Eq. (3); where $\mathbf{R}_{nd}^m = \mathbf{R}_{nd} + \mathbf{r}_m$, $\mathbf{R}_{nd} = n_1 d_1 \mathbf{a}_1 + n_2 d_2 \mathbf{a}_2 + n_3 d_3 \mathbf{a}_3$ and $\tilde{\mathbf{u}}_m$ is the average displacement of atom $m$ at position $\mathbf{R}_{nd}$.

### Reporting summary

Further information on research design is available in the Nature Portfolio Reporting Summary linked to this article.

## Data availability

The data underpinning the findings of this study are available from M.C.N upon reasonable request.

## Code availability

The computational analysis in this study utilized two distinct code-bases. The code for the concurrent phase retrieval algorithm is part of "Bonsu: The Interactive Phase Retrieval Suite" software package, and can be downloaded through PyPI (https://pypi.org/project/Bonsu/) or GitHub (https://github.com/bonsudev/bonsu). Additionally, the code used for the simulations conducted in this study is available within the DiffSim package through PyPI (https://pypi.org/project/diffsim/) or GitHub (https://github.com/bonsudev/diffsim). Both are integral to the research and are made available to ensure transparency and reproducibility in scientific research. The provided links offer direct access to the respective code repositories, enabling other researchers to utilize and build upon the computational work conducted in this study.

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

## Acknowledgements

This work was supported by UK Research and Innovation (UKRI) grant MR/T019638/1 for the University of Southampton Department of Physics & Astronomy. The authors acknowledge the use of the IRIDIS High-Performance Computing Facility, and associated support services at the University of Southampton, in the completion of this work. We acknowledge Diamond Light Source for time on Beamline I16 under Proposal MM30073-1.

## Author contributions

A.H.M. wrote the manuscript with input from all authors. M.C.N. designed and supervised the study. F.L. and N.A.S. prepared and supplied the bulk samples. A.H.M. synthesized the nanocrystals. A.H.M., D.S., M.C.N., D.G.P., S.P.C. and A.B. contributed to the BCDI experiment at D.L.S. A.H.M. and M.C.N. performed data reconstruction and analysis.

## Competing interests

The authors declare no competing interests.
