## [Peer Review File · Nature Communications]

Three-Dimensional Domain Identification in a Single Hexagonal Manganite NanocrystalREVIEWER COMMENTS

Reviewer #1 (Remarks to the Author):

The authors performed the BCDI technique concurrently visualizing the phase image from multiple Bragg reflections to identify the domain of YMnO₃ single crystals. Since the visualization of the unique domain of YMnO₃ is a very significant and spectacular link between condensed matter physics and astrophysics, this work has both originality and universality that can be widely accepted. However, the coarse-grained arguments are left out of the excellent research background; in particular, there is a lack of quantitative and domain visualization evidence. If the following comments are not addressed, publication in Nature Communications will be difficult.

Line 86, p. 2

The equation $\varphi = \mathbf{Q} \cdot \mathbf{u}$ is not explained. The phase is a projection of the displacement field onto the scattering vector, which would be obvious if they knew the BCDI method, but would not make sense to the general reader.

Lines 103-104, p. 3

What is the treatment of the method indicated by "The real space amplitude was updated..."? It appears to be a procedure to proceed with the analysis for different reflections simultaneously, however, it is not clear. In general, BCDI is a method applied to a single reflection, and applying it to multiple reflection spots will result in errors in most cases (e.g., Phys. Rev. Materials. 4, 106001 (2020)).

Furthermore, ferroelectrics commonly studied, such as BaTiO₃, have an anisotropic c/a ratio of about 1.01, and thus pairs of Bragg reflections corresponding to a domain (e.g., 110/101 or 200/002) are often well separated in reciprocal space (e.g., Phys. Rev. Materials. 4, 106001 (2020), Jpn. J. Appl. Phys. 61 SN1008 (2022)); however, the separation is one aspect that makes image reconstruction in BCDI difficult. In the case of YMnO₃, it should be mentioned whether the Bragg reflection pairs corresponding to the domains can withstand the analysis in the reciprocal space.

Line 112, p. 4

The treatment of defects created after grinding is partly unclear; the quality of the sample would be greatly affected by the different conditions here.

Was PLD done under heat? And was annealing done under the atmosphere? It is conceivable that the combination of ultrahigh vacuum and heating could easily make oxygen vacancies.

Lines 117 and 126, p. 4

A quantitative description of the phase is missing. There is little convincing material to say that the phase contrast shown here is a domain boundary. How, for example, can the possibility of dislocation be dismissed?

The phase values are used later in the discussion as averaged, and since the spatial resolution is also derived, it is possible to describe how much the phase has changed (not just higher or lower) per 13 nm.

Line 122, p. 4

If the topological defects are preserved, wouldn't the vortex remain throughout the sample even if the sample is crushed? Is it possible to extract only a portion of the vortex as a

system?

Line 128, p. 4

From where can we determine that the phase contrast in the result of (110) is "small crystal defects"? I think (110) is a typo for (111).

Line 133, p. 4

A reference should be given for "which has been reported in other studies of this material."

Line 141, p. 5

Like ϵ_{yy} , the fine structures of ϵ_{xz} and ϵ_{zz} are also consistent with the (111) phase map, but why no mention of this structure (which is probably not a ferroelectric domain, though)?

Figure 4, p. 6

Since the c-vector is drawn flat, the c-axis direction is not identified. It would be better to use a three-dimensional c-vector.

Line 172, P.7

The authors claim $[\gamma, \beta+]$ as the domain configuration, but I would like to know the logic to reject the phase difference of $[\alpha+, \gamma-]$.

Also, the sign of the phase can be inverted due to the Friedel pair relation. The authors are discussing only a single reflection (212) here, and it is possible that we are looking at $-Q$ against a hypothetical Q vector; i.e., the possibility that the circular mean is -1.5 instead of 1.5 . How is this possibility ruled out?

Line 173, p. 8

I don't understand why a positive metric would make anti-vortex. If you are talking about Higgs-Goldstone physics, it should be described as such.

Line 180, p. 8.

"This is consistent with the experimental results as a similar constant non-zero phase has been observed." refers to which of the present results? Or does it refer to a previous study?

Reviewer #2 (Remarks to the Author):

The manuscript "Three-Dimensional Domain Identification in a Single Hexagonal Manganite Nanocrystal" by Mokhtar et al is an exciting application of the nanometer strain imaging technique Bragg coherent diffraction imaging (BCDI) to a unique topological multiferroic material YMnO_3 . Strengths of this paper is that it utilizes measurements of multiple Bragg peaks to reconstruct a high resolution (x) three-dimensional strain map of the material and has accompanying atomistic modeling. BCDI enables visualization of the strain domains in this nanocrystal which is exciting since understanding of the domain wall structure and motion is critical to controlling the potentially useful applications of ferroelectrics. Additionally, the Mexican hat potential structure has been theorized as a potential analog for cosmic theories, making this work additionally impactful and interesting for a wide audience. Finally, it should be noted that is this a tremendously difficult experiment to conduct on a functional material such as a multiferroic. All of these strengths make the article very suitable for the audience of Nature Communication, however there are several major concerns I have and many minor ones that should be addressed before it should be published. I describe these below:

Major concerns:

- 1) One major concern is what evidence is there that all five peaks come from the same nanocrystal. Are there any SEM images of the nanocrystals so we can see their shape, typical size and density? If the nanocrystals are very regular in shape, is it not possible the peaks could come from separate crystals? There is a lack of description of the sample which makes the reader concerned about the validity of the results. However, additional information on the sample would greatly strengthen the article.
- 2) There is a lack of experimental information that needs to be added to the article. For example, what is the sample to detector distance and what is the detector used? What was the approximate flux into the focused spot? These are important parameters to be discussed.
- 3) The description of the comparison of the extracted displacement and phase and the simulated Bragg peaks and phase was not clear (i.e. Figure 5 and surrounding discussion). This should be revised or expanded upon as it is critical to the demonstration of the Mexican hat potential.
- 4) How would domain and polarization of this relatively small nanocrystal compare to bulk properties? Do you see any of evidence of the topologically protected surface state mentioned in the abstract. This should be discussed.

Other items:

- 1) On page 2, line 50, you say 'prototype yttrium manganate' but would 'prototypical yttrium manganate' be a better phrase?
- 2) I believe when figures and equations are referenced (i.e. 'figure 1a') that Figure and Equation should be capitalized.
- 3) On page 2, lines 85-86, u and ϕ should be defined for clarity of reading (i.e. u should appear after 'atomic displacements' and ϕ should be defined as the exit wave phase).
- 4) You do not reference your own work on multi-Bragg peak reconstructions when you discuss it at the bottom of page 2, or others in the field such as Gao et al Phys. Rev. B 103, 014102 (2021) and Wilkin et al Phys. Rev. B 103 (21): 214103. These should probably be referenced in that part of the discussion. Additional information should also be included about the reconstruction recipe (how many iterations, how was the coupling between different peaks accomplished, etc).
- 5) I do not believe the 3 Bragg peaks measured (111, 110, and 212) are orthogonal. How then can you obtain a full 3D strain tensor?→ This should be described.
- 6) How can you explain the separate floating densities outside of the main bulk of the nanocrystal in Figure 3? These seems like a poor support constraint, surely there are not atoms floating outside the main nanocrystal. What is the isosurface density level of Figure 3?
- 7) Also, in Figure 3, where is the substrate in comparison to the nanocrystal?
- 8) How can you be assured that the PLD added a 5-10 nm layer only?
- 9) Please reference what the vortex core density is from Reference 29 so readers do not need to go look it up.
- 10) On page 4, lines 133-134, you state "The bulk crystal shows no obvious signs of Oxygen depletion at the surface when it is scaled down to the nanoscale." But you do not provide any evidence of this. How to you come to this conclusion? This should be addressed in the text.
- 11) On page 8, line 179 you say 'doesn't'. This should probably be changed to 'does not' to make it more professional.
- 12) Cannot neutron scattering of Manganates provide some idea of their domain structure in the bulk?
- 13) Is 6 hours of annealing sufficient to remove any residual strain of the grinding and PLD?

For high quality gold nanocrystal typically used in BCDI, much higher temperatures (900 C) and longer periods are necessary (upwards of 24 hours).

14) Reference 14 seems to be in preparation, is this allowed in Nat. Comm.? Would an Arxiv reference be more appropriate if possible?

Reviewers' Comments and Responses for: Three-Dimensional Domain Identification in a Single Hexagonal Manganite Nanocrystal

Ahmed H. Mokhtar^{1*}, David Serban¹, Daniel G. Porter³, Frank Lichtenberg², Stephen P. Collins³, Alessandro Bombardi³, Nicola A. Spaldin², Marcus C. Newton¹

¹ School of Physics and Astronomy, University of Southampton

² Department of Materials, ETH Zurich

³ Beamline I16, Diamond Light Source

January 24, 2024

Dear Reviewers,

I would like to extend my sincere gratitude for your insightful comments and suggestions. I hope that the responses provided comprehensively address your queries. In response to one of the comments, we have refined the reconstruction process to yield a more compact object; this adjustment did not significantly influence the observed domain structure. A major change implemented in our methodology involved rotating the crystal such that its c-axis aligns with the z-axis. This adjustment facilitates a clearer interpretation of the results, particularly in relation to the strain plots. Consequently, there are notable modifications in Figures 3 and 4 reflecting these changes. Additionally, in response to some of the comments, I have highlighted in blue text the additions made to the manuscript for easier identification.

Sincerely,

Ahmed H. Mokhtar

Reviewer 1

Comment 1): The authors performed the BCDI technique concurrently visualizing the phase image from multiple Bragg reflections to identify the domain of YMnO₃ single crystals. Since the visualization of the unique domain of YMnO₃ is a very significant and spectacular link between condensed matter physics and astrophysics, this work has both originality and universality that can be widely accepted. However, the coarse-grained arguments are left out of the excellent research background; in particular, there is a lack of quantitative and domain visualization evidence. If the following comments are not addressed, publication in Nature Communications will be difficult.

Comment 2): Line 86, p. 2 The equation $\phi = \mathbf{Q} \cdot \mathbf{u}$ is not explained. The phase is a projection of the displacement field onto the scattering vector, which would be obvious if they knew the BCDI method, but would not make sense to the general reader.

- Thank you for your valuable feedback regarding the explanation of the equation $\phi = \mathbf{Q} \cdot \mathbf{u}$ in our manuscript. We acknowledge that a clearer description is necessary for readers who might not be familiar with Bragg Coherent Diffractive Imaging (BCDI) methods. In response to your comment, we have revised the relevant section of our manuscript to include a more comprehensive explanation of this equation and its significance in the context of our study. In the revised manuscript, we explain that the phase ϕ is directly proportional to the vector displacement field, $\mathbf{u}(\mathbf{r})$, of the atoms from their ideal lattice points, and the scattering vector \mathbf{Q} . This relationship implies that the phase distribution in the BCDI image at a specific Bragg peak is equal to the scalar product of the displacement field and the scattering vector of the measured Bragg peak. This detailed description is now accompanied by references to authoritative sources that elaborate on this principle of BCDI:

We believe that this revised explanation, along with the added references, will not only address your concern but also enhance the manuscript's clarity for all readers.

- The reconstructed real space phase information, $\phi(\mathbf{r})$, reveals a projection of the atomic displacement, $\mathbf{u}(\mathbf{r})$, at a point \mathbf{r} in the crystal from equilibrium along the direction of the chosen \mathbf{Q} -vector in the experiment, according to $\phi(\mathbf{r}) = \mathbf{Q} \cdot \mathbf{u}(\mathbf{r})$. Hence, the real space phase enables the retrieval of atomic

displacement parallel to the Bragg peak reciprocal lattice vector. The derivative of this displacement along the same vector provides the normal strain field, which serves as a proxy for identifying structural changes within the material. However, for a single Bragg peak, only a single projection of the displacement field is obtained. Consequently, components of the displacement field perpendicular to the scattering vector \mathbf{Q} will not be observed in the BCDI measurement. To recover the full displacement field and strain tensor, diffraction patterns from multiple non-planar \mathbf{Q} -vector directions are required.

Comment 3): Lines 103-104, p. 3 What is the treatment of the method indicated by "The real space amplitude was updated..."? It appears to be a procedure to proceed with the analysis for different reflections simultaneously, however, it is not clear. In general, BCDI is a method applied to a single reflection, and applying it to multiple reflection spots will result in errors in most cases (e.g., Phys. Rev. Materials. 4, 106001 (2020)).

- Thank you for your insightful comment on the treatment of the concurrent phase retrieval method in our manuscript. In response, we have expanded the Methods section to include a detailed description of the Multi-Reflection Bragg Coherent Diffractive Imaging (BCDI) approach utilized in our study. We acknowledge that while BCDI is traditionally applied to single reflections, advancements in the field have led to the development of Multi-reflection BCDI. This method allows for the concurrent analysis of multiple Bragg peaks from a single nanoparticle, effectively reconstructing vector-valued lattice distortion fields within nanoscale crystals. Our study implements this approach as demonstrated in recent publications. Notably, a paper published in npj Computational Materials demonstrates this method's effectiveness in reconstructing lattice distortion fields from multiple Bragg reflections, including challenging cases with discontinuities in the lattice distortion fields.

Maddali, S., Frazer, T.D., Deegan, N. et al. Concurrent multi-peak Bragg coherent x-ray diffraction imaging of 3D nanocrystal lattice displacement via global optimization. npj Comput Mater 9, 77 (2023). <https://doi.org/10.1038/s41524-023-01022-7>

Yuan Gao, Xiaojing Huang, Hanfei Yan, and Garth J. Williams Bragg coherent diffraction imaging by simultaneous reconstruction of multiple diffraction peaks. Phys. Rev. B 103, 014102 – Published 8 January 2021

We believe that the inclusion of this expanded description and reference to successful implementations in the field will address your concerns and further substantiate the methodology used in our study.

- Reconstructions of the full displacement field were performed concurrently using *The Interactive Phase Retrieval Suite* [34] as described in Ref. [30] by a combination of Fienup's HIO Mask [35] and Error Reduction [27] algorithms, with the support created using a manual version of the shrink wrap method. This innovative approach allows for the concurrent analysis of multiple Bragg peaks from a single nanoparticle, thereby enabling the effective reconstruction of vector-valued lattice distortion fields within the nanocrystal. It accounts for geometric factors arising in BCDI measurements and allows for a globally constrained single image reconstruction to multiple Bragg peak measurements.

Additionally, our method involved an additional real space constraint based on the mean of the different amplitudes concurrently reconstructed from multiple Bragg reflections, since these reflections originated from the same crystal. This step was instrumental in accelerating the convergence of our phase retrieval process, enhancing the accuracy and efficiency of our reconstruction of the 3D strain field

Comment 4): Furthermore, ferroelectrics commonly studied, such as BaTiO₃, have an anisotropic c/a ratio of about 1.01, and thus pairs of Bragg reflections corresponding to a domain (e.g., 110/101 or 200/002) are often well separated in reciprocal space (e.g., Phys. Rev. Materials. 4, 106001 (2020), Jpn. J. Appl. Phys. 61 SN1008 (2022)); however, the separation is one aspect that makes image reconstruction in BCDI difficult. In the case of YMnO₃, it should be mentioned whether the Bragg reflection pairs corresponding to the domains can withstand the analysis in the reciprocal space.

- Thank you for your comment regarding the analysis of Bragg reflection pairs in YMnO₃ and the challenges associated with image reconstruction in BCDI, particularly in relation to the separation of these pairs in reciprocal space.

In our study, we specifically addressed this aspect for YMnO₃. Our simulated atomistic results confirm that the ferroelectric domains in YMnO₃ are indeed encoded in the same diffraction pattern. This is because the c/a ratio does not differ in different domains, only the origin of rotation of the MnO₅ polyhedra differs. This is a crucial distinction from commonly studied ferroelectrics like BaTiO₃, where the anisotropic c/a ratio leads to a well-separated pair of Bragg reflections corresponding to a domain in reciprocal space.

To provide a comprehensive understanding of our approach and findings, we have included in the Code Availability statement a link to the GitHub repository where our simulation scripts are available. This will allow for a detailed examination of our methodology and the basis for our conclusions.

We believe that these additions will clarify the methodological aspects unique to our study of YMnO₃ and further substantiate the robustness of our analysis in reciprocal space.

Comment 5): Line 112, p. 4 The treatment of defects created after grinding is partly unclear; the quality of the sample would be greatly affected by the different conditions here. Was PLD done under heat? And was annealing done under the atmosphere? It is conceivable that the combination of ultrahigh vacuum and heating could easily make oxygen vacancies.

- Thank you for your insightful inquiry about the treatment of defects post-grinding and the conditions under which Pulsed Laser Deposition (PLD) was conducted in our study.

To clarify, both the PLD and the subsequent annealing processes were carried out under vacuum conditions. This approach was chosen to maintain the stoichiometry of the YMnO₃ crystals and minimize the introduction of contaminants, which could arise from low vacuum combined with heating.

In our study, the precision of our diffraction measurements is highlighted by the close alignment of observed and calculated 2θ values. This alignment is quantitatively expressed as $\sum \frac{|2\theta_{\text{obs}} - 2\theta_{\text{calc}}|}{2\theta_{\text{calc}}} < 10^{-3}$, indicating a high level of accuracy. This correspondence confirms the correct phase and stoichiometry of the material, suggesting that any defects or variations introduced post-grinding did not significantly alter the fundamental structure.

- The precision in our diffraction measurements is underscored by the close match between the observed 2θ values and their calculated counterparts for the studied reflections. This match is quantitatively expressed by the quantity $\sum \frac{|2\theta_{\text{obs}} - 2\theta_{\text{calc}}|}{2\theta_{\text{calc}}} < 10^{-3}$ indicating an exceptionally high level of accuracy. This correspondence not only confirms the accuracy of our measurements but also validates the correct phase and stoichiometry of the material.

Comment 6): Lines 117 and 126, p. 4 A quantitative description of the phase is missing. There is little convincing material to say that the phase contrast shown here is a domain boundary. How, for example, can the possibility of dislocation be dismissed? The phase values are used later in the discussion as averaged, and since the spatial resolution is also derived, it is possible to describe how much the phase has changed (not just higher or lower) per 13 nm.

- Thank you for your valuable comment requesting a quantitative description of the phase contrast and clarification on its interpretation as a domain boundary in our study.

We have conducted a focused analysis on the phase values at the domain boundaries, measuring the specific phase change across these boundaries with a resolution of 13 nm. This analysis enables us to quantify the phase shifts in precise terms, thereby providing a clearer understanding of the domain boundary characteristics.

We believe these additions to our manuscript offer a more comprehensive and quantitative understanding of the phase changes observed, addressing your concerns effectively.

In the case of a dislocation, the phase contrast typically exhibits a significant phase difference localized at the dislocation site, followed by a gradual increase or decrease to zero along the length of the crystal. This is caused by lattice relaxation away from the dislocation edge. This is in contrast to the phase changes associated with domain boundaries, which are characterized by a step-like change in phase.

The paper by Cherukara et al. from Nature Communications presents 3D X-ray imaging of strain fields around a screw dislocation, showing a circular wrap and sharp strain discontinuity. This differs from the step-like phase changes we associate with domain boundaries, which is observed in our results.

Cherukara, M.J., Pokharel, R., O'Leary, T.S. et al. Three-dimensional X-ray diffraction imaging of dislocations in polycrystalline metals under tensile loading. *Nat Commun* 9, 3776 (2018). <https://doi.org/10.1038/s41467-018-06166-5>

Additionally, in our work, the phase contrast is predominantly visible in the (212) direction, and notably absent in the (110) and (111) phase maps. This selective visibility supports the interpretation that (212) contrasts are indicative of domain boundaries rather than dislocations, as a dislocation would likely be visible in the other phase maps.

This selective visibility, along with the symmetry in the (111) direction that aligns with our atomistic simulations, suggests these contrasts are indicative of domain boundaries, not dislocations.

This evidence, combining direct observation and computational modeling, underpins our conclusion that the phase contrast observed is representative of domain boundaries, not dislocations.

- The phase map along (212) reveals the existence of two domains with high phase values and hence large displacement along that \mathbf{Q} vector direction, which corresponds to the ferroelectric domains in the material. Domain 1 exhibits a mean phase value of $2.25 \pm 0.006\text{rad}$, while Domain 2 has a mean phase value of $-2.54 \pm 0.004\text{rad}$.

In contrast to dislocations, which typically show significant phase differences localized at the dislocation site and a gradual phase change away from the dislocation site due to lattice relaxation, the phase changes observed here are step-like, indicative of domain boundaries. A dislocation typically demonstrates circular wraps and sharp strain discontinuities, differing from our step-like phase changes.

The low standard error of the phase is indicative small phase gradient in each domain, hence we do not have the gradual relaxation indicative of a dislocation.

Additionally, in our work, the phase contrast is predominantly visible in the (212) direction, and notably absent in the (110) and (111) phase maps. This selective visibility supports the interpretation that (212) contrasts are indicative of domain boundaries rather than dislocations, as a dislocation would likely be visible in the other phase maps. Additionally, the selective visibility is consistent with our atomistic simulations discussed later in the paper.

Comment 8): Line 122, p. 4 If the topological defects are preserved, wouldn't the vortex remain throughout the sample even if the sample is crushed? Is it possible to extract only a portion of the vortex as a system?

- Thank you for your query regarding the preservation of topological defects and the extraction of vortex portions in our samples.

In our study, the topological defects within the growth crystal are indeed preserved. When the crystal is subsequently cut, the portion obtained still contains a part of the original domain structure, which includes these topological defects. As for the vortex, in our case, it is larger than the sample itself. Hence, while the entire vortex structure may extend beyond the sample boundaries, the section within our sample still provides valuable insights into the domain structure and topological characteristics. Moreover, it also shows that the size of the topological structure is not affected by the size of the crystal, i.e. we do not observe a change in the size of the vortex structure as a consequence of the size of the crystal. This is to be expected as the structure is topologically protected.

This aspect of our methodology ensures that, despite the physical limitations of sample size, the fundamental properties of the topological defects are retained and observable within our analysis.

- Piezoforce microscopy (PFM) images of the bulk crystal in Reference [29] reveal the density of vortex cores at the surface to be $0.3 \frac{\text{cores}}{\mu\text{m}^2}$. Given this vortex density and the small size of the crystal, observation of a complete vortex or multiple vortices is improbable. Therefore, we are observing a small portion of the full structure, a section that contains two domains separated by a domain wall.

Our investigations reveal that the vortex structure within the crystal is independent of the crystal's size as the topological structures remain unaffected by variations in the crystal's dimensions. Notably, the stability and consistency of the vortex size, irrespective of the crystal size, align with theoretical expectations. Such behavior underscores the topological protection inherent in these structures, suggesting that their characteristics are fundamentally governed by topological constraints rather than geometrical dimensions of the crystal.

Comment 9): Line 128, p. 4 From where can we determine that the phase contrast in the result of (110) is "small crystal defects"? I think (110) is a typo for (111).

- Thank you for your comment regarding the interpretation of phase contrast in the (110) results of our study.

I would like to clarify that the reference to (110) is intentional and not a typo. In our analysis, the (110) \mathbf{Q} -vector direction does not have any component in the direction of the c -axis. Therefore, it does not detect yttrium ion displacements. Our simulations indicate that a perfect crystal without defects should exhibit a zero phase along the 110 direction. Thus, the observed phase in our results is attributed to small crystal defects or secondary strain effects.

Comment 10): Line 133, p. 4 A reference should be given for "which has been reported in other studies of this material."

- Thank you for your comment regarding the need for a reference on other studies of the material. The study we refer to is related to the bulk material preparation conducted at ETH Zurich, which demonstrates a stoichiometric phase of YMnO_3 without evident oxygen diffusion (reference [13] in our manuscript).

Additionally, the lack of varied phase contrasts at the surface in our results further supports the absence of significant oxygen diffusion. We have updated the manuscript to include this reference and to clarify our findings more accurately.

- Additionally, the crystal shows no obvious signs of Oxygen depletion at the surface when scaled down to the nanoscale. The absence of Oxygen diffusion is consistent with the bulk material. Typically, Oxygen depletion would manifest as a distinct phase at the crystal's surface, but this is not observed in our results, further supporting the integrity of our material preparation.

Comment 11): Line 141, p. 5 Like ϵ_{yy} , the fine structures of ϵ_{xz} and ϵ_{zz} are also consistent with the (111) phase map, but why no mention of this structure (which is probably not a ferroelectric domain, though)?

- Thank you for your insightful query about the fine structures of the ϵ_{xz} and ϵ_{zz} components in relation to the (111) phase map.

In our study, the (111) direction does not prominently show the ferroelectric domains. This is consistent with both our simulations and experimental results. The tensor directions along the c-axis exhibit the strongest contrast or strain boundary, which is where we predominantly observe the ferroelectric domains. However, slight deviations from the c-axis could be observed in orthogonal directions due to secondary effects, such as strain that breaks the symmetry, revealing some structure in these directions.

In response to your comment, we have expanded our manuscript to address this point. We have rotated the crystal such that the c-axis aligns with the z-axis to isolate the effect in one direction.

We added that the structure resembling the (111) phase map is visible in multiple components which is likely caused by secondary strain effects. We have revised the manuscript to reflect these insights and emphasize our focus on the order parameter changes along the c-axis, which is central to our atomistic simulations.

It's also important to note that real systems may not follow theoretical models exactly. Imperfections in the crystal can cause less pronounced variations, leading to the observations in ϵ_{xz} and ϵ_{xx} . These subtler features are likely a result of the complex interplay of strain and crystal imperfections.

- The domain wall is most evident in the ϵ_{zz} components as illustrated in Figure 4. It is marked by distinct purple strings and is notably absent in the ϵ_{xx} and ϵ_{yy} components. The strings align with the domain wall identified in the (212) phase map, highlighting the displacement direction between different domains. The c-axis vector aligns with the z-axis which suggests that the observed displacement between domains is primarily due to Yttrium ion movement along the c-axis.

In the strain tensor, dislocations typically exhibit circular wraps and a profile of gradual increase or decrease. The observed profile further supports the existence of ferroelectric domains.

Comment 12): Figure 4, p. 6 Since the c-vector is drawn flat, the c-axis direction is not identified. It would be better to use a three-dimensional c-vector.

- Thank you for your suggestion regarding the depiction of the c-vector in Figure 4 of our manuscript.

In response to your feedback, we enhanced the figure by including a 3D representation of the c-axis vector on the rendering of the crystal, the c-axis is now along the z-direction which is represented in Figures 3b and 3c. This adjustment provides a clearer and more accurate visualization of the c-axis direction in relation to the crystal structure. While our current figure shows only a component of the c-vector, the updated 3D representation will offer a more comprehensive view, aiding in the interpretation of the crystal orientation and its properties.

We appreciate your suggestion and believe this enhancement will significantly improve the clarity of our visual representation in the manuscript.

Comment 13): Line 172, P.7 The authors claim $[\gamma-, \beta+]$ as the domain configuration, but I would like to know the logic to reject the phase difference of $[\alpha+, \gamma-]$. Also, the sign of the phase can be inverted due to the Friedel pair relation. The authors are discussing only a single reflection (212) here, and it is possible that we are looking at -Q against a hypothetical Q vector; i.e., the possibility that the circular mean is -1.5 instead of 1.5. How is this possibility ruled out?

- Thank you for your inquiry about the selection of the $[\gamma-, \beta+]$ domain configuration over the $[\alpha+, \gamma-]$ configuration in our analysis.

To determine the most likely domain configuration, we computed the error between these two possibilities and found that the $[\gamma-, \beta+]$ configuration aligns more closely with both our experimental data and theoretical expectations. This configuration demonstrated a better alignment with the theory and a lower error margin, leading us to identify it as the most likely configuration.

Regarding your question about the possibility of observing the $-Q$ vector instead of Q in our single reflection (212) analysis, this was addressed by examining the magnitude of the circular mean instead, yielding that the $[\gamma-, \beta+]$ configuration aligns the closest. Additionally, we analyzed the relative angles between the individual q -vector directions, which further supported our conclusion that we are observing the Q vector, not the $-Q$. The relative angular analysis, supported by our computational code, shows consistency with the expected Q -vector orientation, as the scenario of observing $-Q$ for all vectors highly unlikely. This approach allowed us to confidently determine the correct pair.

We believe these methods provide a robust basis for our configuration choice and address the potential concerns regarding vector orientation.

- Within each domain in the simulated data, the phase value is constant, however, in the experimental results, an average was computed for each domain. Figure 5c displays the result from these computations clearly showing a close match of the experimental and simulated circular mean at the $\gamma-, \beta+$ and $\alpha+, \gamma-$, regions with a χ^2 of 0.007 and 0.071, respectively. This signifies that the $\gamma-, \beta+$ forms a better fit with a χ^2 value that is an order of magnitude higher. It can then be determined that the two observed domains in the (212) phase map are $\beta+$ and $\gamma-$ for the positive and negative phase regions, respectively, as they are better alignment with the theoretical model.

Comment 14): Line 173, p. 8 I don't understand why a positive metric would make anti-vortex. If you are talking about Higgs-Goldstone physics, it should be described as such.

- Thank you for your comment regarding the interpretation of a positive metric in relation to the anti-vortex configuration.

In our study, the determination of the vortex or anti-vortex configuration is based on the analysis of the circular mean of the simulated phase, with careful consideration of the order. If the signs of the phase values match, it indicates the same order, consistent with an anti-vortex configuration. Conversely, if the signs were different, it would imply a reversal in order, leading to a vortex configuration. Our discussion in this context does not directly engage with Higgs-Goldstone physics but focuses specifically on the phase order in our simulation results.

- The circular mean is a non-commutative quantity, hence, its polarity has some significance in determining the vortex direction. The circular mean values in Figure 5c for the simulated results were done with an anti-vortex order, hence, a matched polarity with the experimental results is indicative that the two domains observed are part of an anti-vortex structure in this case, i.e. a clockwise movement around the brim of the Mexican-hat potential as illustrated in Figure 5d.

Comment 15): Line 180, p. 8. "This is consistent with the experimental results as a similar constant non-zero phase has been observed." refers to which of the present results? Or does it refer to a previous study?

- Thank you for seeking clarification on the statement. The reference to "a similar constant non-zero phase" pertains specifically to our (111) experimental results. It's important to clarify that while the phase observed in these results is not strictly constant, it is characteristically noisy and consistently non-zero. This observation aligns with the absence of the domain primary order parameter in our analysis. We have revised the wording in the manuscript to more accurately reflect this.
- The simulated results of the (111) reflection demonstrated a constant non-zero phase in the whole crystal, i.e. the domains are indistinguishable from each other along that direction. It can be said that the projection of the Yttrium ions' displacements along that Q -vector direction poses symmetry that detects the displacements, but does not distinguish them from each other. This result is consistent with the experimental result for the (111) phase map as we observe, although not strictly uniform, a non-zero phase in addition to the domains being indistinguishable from each other. The absence of the domain primary order parameter within our the experimental and the simulated results further highlights the discussed symmetry.

Reviewer 2

Remarks to the Author:

The manuscript “Three-Dimensional Domain Identification in a Single Hexagonal Manganite Nanocrystal” by Mokhtar et al is an exciting application of the nanometer strain imaging technique Bragg coherent diffraction imaging (BCDI) to a unique topological multiferroic material YMnO₃. Strengths of this paper is that it utilizes measurements of multiple Bragg peaks to reconstruct a high resolution (x) three-dimensional strain map of the material and has accompanying atomistic modeling. BCDI enables visualization of the strain domains in this nanocrystal which is exciting since understanding of the domain wall structure and motion is critical to controlling the potentially useful applications of ferroelectrics. Additionally, the Mexican hat potential structure has been theorized as a potential analog for cosmic theories, making this work additionally impactful and interesting for a wide audience. Finally, it should be noted that is this a tremendously difficult experiment to conduct on a functional material such as a multiferroic. All of these strengths make the article very suitable for the audience of Nature Communication, however there are several major concerns I have and many minor ones that should be addressed before it should be published. I describe these below:

Major concerns:

Comment 1): One major concern is what evidence is there that all five peaks come from the same nanocrystal. Are there any SEM images of the nanocrystals so we can see their shape, typical size and density? If the nanocrystals are very regular in shape, is it not possible the peaks could come from separate crystals? There is a lack of description of the sample which makes the reader concerned about the validity of the results. However, additional information on the sample would greatly strengthen the article.

- Thank you for your insightful comment regarding the evidence for all five peaks originating from the same nanocrystal.

In our manuscript, we now clarify that the nanocrystals are irregular in shape and size, which is a direct result of the mechanical grinding process used in their preparation. This method inherently produces a random distribution in size and morphology, making it unlikely for separate crystals to consistently exhibit similar diffraction patterns.

Additionally, we have performed an in-depth analysis of the fringe frequency and shape within our Bragg coherent diffraction imaging data. This analysis confirms that the diffraction patterns indeed originate from the same nanocrystal, substantiating the validity of our results. This information is now reflected in the main manuscript and the fringe frequency analysis diagram is available in the supplementary material, Figure S2.

Comment 2): There is a lack of experimental information that needs to be added to the article. For example, what is the sample-to-detector distance and what is the detector used? What was the approximate flux into the focused spot? These are important parameters to be discussed.

- Thank you for highlighting the need for more detailed experimental information in our manuscript.

We agree that details such as the sample-to-detector distance, the type of detector used, and the approximate flux into the focused spot are crucial for a comprehensive understanding of our experimental setup. To address this, we will include these specifics in the supplementary material of our manuscript. This addition will ensure that all relevant experimental parameters are transparently communicated and accessible to readers, further enhancing the clarity and reproducibility of our study.

- The detector employed in our experiments was the Quad Merlin detector, featuring 515×515 pixels, with each pixel measuring $55\mu\text{m} \times 55\mu\text{m}$. The sample-to-detector distance during measurements was 1.31 meters. The full beam x-ray flux at the sample is 3×10^{13} photons/s with a beam size of $200\mu\text{m}(h) \times 30\mu\text{m}(v)$, for these measurements we employed slits before the sample with size $20\mu\text{m} \times 20\mu\text{m}$, reducing the beam size and used Al foils to attenuate the incident flux by 70% to mitigate beam damage to the sample.

Comment 3): The description of the comparison of the extracted displacement and phase and the simulated Bragg peaks and phase was not clear (i.e. Figure 5 and surrounding discussion). This should be revised or expanded upon as it is critical to the demonstration of the Mexican hat potential.

- Thank you for your feedback regarding the description and comparison of the extracted displacement and phase with the simulated Bragg peaks and phase, particularly in relation to Figure 5 and the discussion surrounding the demonstration of the Mexican hat potential.

We acknowledge the need for clarity in this critical part of our manuscript. While it may not be feasible to apply the suggestions directly without further specifics, we have revised the last paragraphs of the relevant section to provide a more detailed explanation of the comparison process.

We have included a description of why a positive circular mean metric indicates an anti-vortex domain structure. We also included chi-squared errors highlighting the alignment of the circular variance of the simulated domain pair with the experimental ones. Additionally, we rephrased some sentences to ensure clarity for the reader.

We hope this revision addresses your concerns and better elucidates the connection between our experimental results and the theoretical Mexican hat potential model.

4) How would domain and polarization of this relatively small nanocrystal compare to bulk properties? Do you see any of evidence of the topologically protected surface state mentioned in the abstract. This should be discussed.

- Thank you for your question regarding the comparison of domain and polarization properties between our studied nanocrystal and bulk materials, as well as the evidence of the topologically protected surface state.

Piezoresponse Force Microscopy (PFM) images in Reference 29 clearly show the full cloverleaf structure in the bulk crystals. In the case of the smaller nanocrystals studied, we observe a fraction of this domain structure. This difference underlines the distinction between nanocrystal and bulk crystal properties. As for the topologically protected state mentioned in the abstract, it refers to the domain walls, which are indeed observed in our study. This observation underscores the relevance of our findings to understanding the topological aspects of domain structures in these materials.

Figure 3d has been included to visualize how the two observed domains form part of the topologically protected structure in the abstract.

- Piezoforce microscopy (PFM) images of the bulk crystal in Reference [29] reveal the density of vortex cores at the surface to be $0.3 \frac{\text{cores}}{\mu\text{m}^2}$. Given this vortex density and the small size of the crystal, observation of a complete vortex or multiple vortices is improbable. Therefore, we are observing a small portion of the full structure, a section that contains two domains separated by a domain wall.

Our investigations reveal that the vortex structure within the crystal is independent of the crystal's size as the topological structures remain unaffected by variations in the crystal's dimensions. Notably, the stability and consistency of the vortex size, irrespective of the crystal size, align with theoretical expectations. Such behavior underscores the topological protection inherent in these structures, suggesting that their characteristics are fundamentally governed by topological constraints rather than geometrical dimensions of the crystal.

Other items:

Comment 1): On page 2, line 50, you say 'prototype yttrium manganate' but would 'prototypical yttrium manganate' be a better phrase?

- Thank you for your suggestion regarding the phrasing on page 2, line 50 of the manuscript. We agree that 'prototypical yttrium manganate' is indeed a more appropriate phrase. The manuscript was revised accordingly to reflect this change.

Comment 2): I believe when figures and equations are referenced (i.e. 'figure 1a') that Figure and Equation should be capitalized.

- Thank you for your attention to detail in the formatting of our manuscript. We agree with your observation and will ensure that all references to figures and equations are properly capitalized, as per standard academic conventions.

Comment 3): On page 2, lines 85-86, u and ϕ should be defined for clarity of reading (i.e. u should appear after 'atomic displacements' and ϕ should be defined as the exit wave phase).

- Thank you for your valuable feedback regarding the clarity of definitions for ' u ' and ' ϕ ' on page 2, lines 85-86 of our manuscript. We will revise this section to clearly define ' u ' as the atomic displacements and ' ϕ ' as the real space complex phase ensuring enhanced clarity and readability.

Comment 4): You do not reference your own work on multi-Bragg peak reconstructions when you discuss it at the bottom of page 2, or others in the field such as Gao et al Phys. Rev. B 103, 014102 (2021) and Wilkin et al Phys. Rev. B 103 (21): 214103. These should probably be referenced in that part of the discussion. Additional information should also be included about the reconstruction recipe (how many iterations, how was the coupling between different peaks accomplished, etc).

- Thank you for pointing out the need for additional references and details in our discussion on multi-Bragg peak reconstructions. We have included references to pertinent works in the field, such as those by Gao et al. (Phys. Rev. B 103, 014102 (2021)) and Wilkin et al. (Phys. Rev. B 103 (21): 214103), to enrich this part of the discussion.

Furthermore, we have expanded the methods section to provide a more detailed description of the reconstruction recipe.

Comment 5): I do not believe the 3 Bragg peaks measured (111, 110, and 212) are orthogonal. How then can you obtain a full 3D strain tensor?— This should be described.

- Thank you for your comment regarding the measurement of the three Bragg peaks (111, 110, and 212) and their relation to the full 3D strain tensor.

It is important to clarify that for obtaining a full 3D strain tensor, the Bragg peaks do not necessarily need to be orthogonal; they need to be non-coplanar. The non-coplanar nature of these Bragg peaks in our experiment allows us to extract the full 3D strain tensor effectively.

- Importantly, the requirement for extracting a full 3D strain tensor is not the orthogonality of these peaks, but their non-coplanarity. The non-coplanar arrangement of the three Bragg peaks in our experimental setup is a key factor that enables the effective extraction of the full 3D strain tensor.

Comment 6): How can you explain the separate floating densities outside of the main bulk of the nanocrystal in Figure 3? These seems like a poor support constraint, surely there are no atoms floating outside the main nanocrystal. What is the isosurface density level of Figure 3?

- Thank you for your observations regarding the floating densities outside the main bulk of the nanocrystal in Figure 3.

The appearance of separate floating densities is not indicative of a poor reconstruction, but rather a consequence of the chosen isosurface representation. In our phase retrieval process, multiple real space constraints derived from the amplitude of different reflections allow for a less stringent support constraint. This sometimes results in non-compact densities that appear outside the main nanocrystal. We have refined the reconstruction process by incorporating more iterations to correct for the floating densities.

However, these additional densities do not influence the phase information within the bulk of the object. We will add the isosurface density level to Figure 3 to provide a clearer understanding of this representation.

Comment 7): Also, in Figure 3, where is the substrate in comparison to the nanocrystal?

- Thank you for inquiring about the substrate's location in relation to the nanocrystal as depicted in Figure 3.

To provide a clearer understanding of the crystal's orientation with respect to the substrate, we will include the q-vector directions in the 3D rendering of the nanocrystal. However, it is important to note that the exact positioning of the crystal relative to the substrate cannot be precisely determined in our setup. Additionally, for the purposes of this study, the specific location of the substrate in relation to the nanocrystal is not critical to the analysis or the conclusions drawn.

Comment 8): How can you be assured that the PLD added a 5-10 nm layer only?

- Thank you for your question regarding the assurance of the 5-10 nm layer thickness in our Pulsed Laser Deposition (PLD) process.

The deposition time and rate used in our experiments were carefully selected based on previous experiments conducted under similar temperature conditions. These prior experiments have consistently shown that the parameters chosen for our PLD process result in a layer thickness within the 5-10 nm range. This empirical evidence gives us confidence in the range of the layer thickness in our study.

Comment 9): Please reference what the vortex core density is from Reference 29 so readers do not need to go look it up.

- Thank you for pointing out the necessity of providing specific details from Reference 29 regarding the vortex core density. In response, we have included the exact vortex core density value from this reference directly in our manuscript. This addition offers readers immediate access to this critical piece of information, enhancing the comprehensiveness and usability of our discussion.

Comment 10): On page 4, lines 133-134, you state “The bulk crystal shows no obvious signs of Oxygen depletion at the surface when it is scaled down to the nanoscale.” But you do not provide any evidence of this. How do you come to this conclusion? This should be addressed in the text.

- Thank you for querying the basis of our conclusion regarding the absence of oxygen depletion in the bulk crystal when scaled down to the nanoscale.

Our conclusion is drawn from the lack of evidence for any phase change typically indicative of oxygen depletion. Such depletion would manifest as a different phase, possibly as a strip around the crystal or in separate regions. Our observations do not show any such signs, which leads us to conclude that oxygen depletion is not a significant factor in our nanoscale samples.

- Additionally, the crystal shows no obvious signs of Oxygen depletion at the surface when scaled down to the nanoscale. Typically, Oxygen depletion would manifest as a distinct phase at the crystal’s surface, but this is not observed in our results, further supporting the integrity of our material preparation.

Comment 11): On page 8, line 179 you say ‘doesn’t’. This should probably be changed to ‘does not’ to make it more professional.

- Thank you for your feedback on the language used in the manuscript. I agree that using ‘does not’ instead of ‘doesn’t’ is more appropriate for a professional scientific document. This change will be made to enhance the formality of the text.

Comment 12): Cannot neutron scattering of Manganates provide some idea of their domain structure in the bulk?

- Thank you for suggesting neutron scattering as a method to investigate the domain structure in manganites. While neutron scattering could potentially provide insights into domain structures, it has limitations compared to Bragg coherent diffraction imaging (BCDI). The neutron scattering cross-section may give us enhanced sensitivity to the MnO distortions, however, it is a flux-limited technique and the current state-of-the-art resolution of neutron imaging is not sufficient to measure domains of this size.

Comment 13): Is 6 hours of annealing sufficient to remove any residual strain of the grinding and PLD? For high quality gold nanocrystal typically used in BCDI, much higher temperatures (900 C) and longer periods are necessary (upwards of 24 hours).

- Thank you for your comment regarding the annealing duration in our study. We acknowledge that longer annealing periods at higher temperatures are typically necessary for high-quality gold nanocrystals in BCDI. However, in the case of our yttrium manganate nanocrystals, prolonged annealing at elevated temperatures risks oxygen depletion due to proximity to the phase transition temperature. Our experimental results, including the absence of surface strain in the reconstructions and corroborating evidence from similar experiments done in the past, suggest that the 6-hour annealing period was sufficient to alleviate the residual strain from grinding and PLD in our specific case.

Comment 14): Reference 14 seems to be in preparation, is this allowed in Nat. Comm.? Would an Arxiv reference be more appropriate if possible?

- Thank you for pointing out the issue with Reference 14. The paper in question has indeed been published. We have updated the reference in our manuscript to accurately reflect its published status, ensuring compliance with the standards of Nature Communications.

REVIEWER COMMENTS

Reviewer #1 (Remarks to the Author):

For reply to comment 8:

The clarification of what "theoretical expectations" are in "Notably, the stability and consistency of the vortex size, irrespective of the crystal size, align with theoretical expectations".

For reply to comment 10:

Unless the properties and structure of the bulk material of this composition are widely known in general, ref.13 should be clearly cited in the appropriate region of the main text.

For reply to comment 13:

The solution obtained by BCDI is either the real image R or its complex conjugate R^* , and it would be impossible to tell whether the solution is R or R^* based on BCDI alone. On that assumption, "we analyzed the relative angles between the individual q -vector directions, which further supports our conclusion that we are observing the Q vector, not the $-Q$ " I didn't quite understand the logic behind this. Can we discriminate R or R^* by analyzing between scattering vectors at several mutually independent reflection indices? On the other hand, the reason why $[\gamma, \beta^+]$ was chosen makes sense.

If the authors are using drawings made with VESTA3, you should cite VESTA3. For example, Fig. 1a looks like it was drawn with VESTA3. Please refer to the VESTA3 Manual for more information.

"Permission to use this software is hereby granted under the following conditions: 1. Drawings produced by VESTA may be used in any publications (original and review articles) provided that its use is explicitly acknowledged. A suitable reference for VESTA is: K. Momma and F. Izumi, "VESTA 3 for three-dimensional visualization of crystal, volumetric and morphology data VESTA 3 for three-dimensional visualization of crystal, volumetric and morphology data," J. Appl. Crystallogr. 44, 1272-1276 (2011). via "https://jp-minerals.org/vesta/archives/VESTA_Manual.pdf"

Reviewer #2 (Remarks to the Author):

The revised manuscript by Mokhtar et al entitled "Three-Dimensional Domain Identification in a Single Hexagonal Manganite Nanocrystal" has been substantially revised and my concerns about it are resolved. I especially appreciate the clarity of the experimental details, reconstructions, and improved figures. The additional text added to the document also improves the clarity, impact, and readability of the article. They present a very convincing argument about the nature of the domain wall. I believe it is now ready to published in Nature Communications. One small item, perhaps in the abstract, it would clearer to state 'Here, we present a multi-peak Bragg coherent x-ray diffraction imaging (BCDI)' by stating multi-peak instead of multi-Bragg.

Reviewers' Comments and Responses for: Three-Dimensional Domain Identification in a Single Hexagonal Manganite Nanocrystal

Ahmed H. Mokhtar^{1*}, David Serban¹, Daniel G. Porter³, Frank Lichtenberg², Stephen P. Collins³, Alessandro Bombardi³, Nicola A. Spaldin², Marcus C. Newton¹

¹ School of Physics and Astronomy, University of Southampton

² Department of Materials, ETH Zurich

³ Beamline I16, Diamond Light Source

March 5, 2024

Dear Reviewers,

I wish to express my deep gratitude for the time and effort you dedicated to the critical analysis of our paper. Your insightful comments and suggestions have significantly contributed to enhancing the robustness and the reader's comprehension of our work.

In the revised manuscript, I have made every effort to address your queries comprehensively.

Sincerely,

Ahmed H. Mokhtar

Reviewer 1

Comment 1:

The clarification of what "theoretical expectations" are in "Notably, the stability and consistency of the vortex size, irrespective of the crystal size, align with theoretical expectations".

Response:

Thank you for your insightful query regarding the clarification of "theoretical expectations" as mentioned in our manuscript. We recognize the importance of explicitly connecting our empirical observations with the established theoretical framework, ensuring a comprehensive understanding of our findings. To clarify, the "theoretical expectations" refer to the principle of topological protection, which ensures that certain properties of the system, such as the size and stability of vortex structures, are preserved against variations in physical dimensions of the crystal. In response to your comment, we have revised the section in question to articulate this more more clearly. The revised passage now reads: "Our investigations reveal that the vortex structure within the crystal is independent of the crystal's size as the topological structures remain unaffected by variations in the crystal's dimensions. This stability and consistency of the vortex size, despite variations in the crystal's dimensions, are anticipated based on the topologically protected nature of these structures, suggesting that their characteristics are fundamentally governed by topological constraints rather than the geometrical dimensions of the crystal." We believe that this clarification provides a clearer context for our discussion on the stability and consistency of vortex sizes. We hope that this revision addresses your concerns.

Comment 2:

Unless the properties and structure of the bulk material of this composition are widely known in general, ref.13 should be clearly cited in the appropriate region of the main text.

Response:

Thank you for your valuable comment. We appreciate your attention to the detail regarding the citation of reference materials in our manuscript. In response to your suggestion, we have carefully reviewed the relevant sections and have now included a clear citation of reference 13, which details the properties and structure of the bulk material in question, within the appropriate section of the main text. This addition should provide our readers with an easy access to the background information essential for the context of our study.

Comment 3:

The solution obtained by BCDI is either the real image R or its complex conjugate R^* , and it would be impossible to tell whether the solution is R or R^* based on BCDI alone. On that assumption, "we analyzed the relative angles between the individual q -vector directions, which further supports our conclusion that we are observing the Q vector, not the $-Q$ " I didn't quite understand the logic behind this. Can we discriminate R

or R^* by analyzing between scattering vectors at several mutually independent reflection indices? On the other hand, the reason why $[\gamma-, \beta+]$ was chosen makes sense.

Response:

To address this, our approach extends beyond exclusive analysis on the Fourier amplitudes. By examining the reconstructed phase in conjunction with the relative angles between the scattering vectors corresponding, we utilise reconstructed the phase information as an additional layer of discrimination between $+\mathbf{Q}$ and $-\mathbf{Q}$, as the crystal structure is not centrosymmetric. Specifically, the simulated data for the (111) reflection indicates a positive phase, contrasting with a negative phase for the $(\bar{1}\bar{1}\bar{1})$ reflection. This phase relationship suggests that the experimental (111) reflection corresponds to a positive \mathbf{Q} direction as it also demonstrated positive phase, leading to the identification of the $+\mathbf{Q}$ rather than its Friedal pair for the experimental (111) reflection.

The rationale behind analyzing multiple reflections and their relative angles stems from the geometrical relationships between different \mathbf{Q} -vectors. For instance, the angle between (111) and $(\bar{2}\bar{1}\bar{2})$ reflections is significantly different from that between (111) and (212), with measured angles of 168.7° and 11.28° , respectively. Our experimental value of the relative angles between these \mathbf{Q} vector matches the latter.

An additional layer of evidence comes from the necessity for consistency among all reflections during concurrent phase retrieval. This process inherently penalizes and filters out inconsistent reflections, which would not reconstruct accurately if the assumed \mathbf{Q} vector orientations were incorrect. The successful reconstruction of our dataset, with all reflections showing consistent phase relations as per our simulations, further validates our methodology and conclusions.

These arguments, in conjunction with the matching magnitude of the circular mean of the (212) reflection with that of the simulated data, provide a robust argument supporting the identification of $+\mathbf{Q}$ vector orientations. There is no matching magnitude of the circular mean if we were observing $-\mathbf{Q}$.

The arguments have now been included in the Supplementary Information (Supplementary 5) of the manuscript for the readers' comprehension of the underlying principles in our analysis.

Comment 4:

If the authors are using drawings made with VESTA3, you should cite VESTA3. For example, Fig. 1a looks like it was drawn with VESTA3. Please refer to the VESTA3 Manual for more information. "Permission to use this software is hereby granted under the following conditions: 1. Drawings produced by VESTA may be used in any publications (original and review articles) provided that its use is explicitly acknowledged. A suitable reference for VESTA is: K. Momma and F. Izumi, "VESTA 3 for three-dimensional visualization of crystal, volumetric and morphology data VESTA 3 for three-dimensional visualization of crystal, volumetric and morphology data," J. Appl. Crystallogr. 44, 1272-1276 (2011). via "<https://jp-minerals.org/vesta/archives/VESTA-Manual.pdf>".

Response:

Thank you for your observation regarding the use of VESTA3 for the creation of the drawings presented in our manuscript. We deeply appreciate your guidance to ensure proper acknowledgment and citation of the software tools that contributed to our research.

In response, we have reviewed our manuscript and have now included the appropriate citation for VESTA3, as recommended. Specifically, we have referenced VESTA in the figure caption to accurately acknowledge the use of VESTA3 for visualization of the crystal structure.

Reviewer 2

Comment 1:

The revised manuscript by Mokhtar et al entitled “Three-Dimensional Domain Identification in a Single Hexagonal Manganite Nanocrystal” has been substantially revised and my concerns about it are resolved. I especially appreciate the clarity of the experimental details, reconstructions, and improved figures. The additional text added to the document also improves the clarity, impact, and readability of the article. They present a very convincing argument about the nature of the domain wall. I believe it is now ready to published in Nature Communications. One small item, perhaps in the abstract, it would clearer to state ‘Here, we present a multi-peak Bragg coherent x-ray diffraction imaging (BCDI)’ by stating multi-peak instead of multi-Bragg.

Response:

We are deeply grateful for your positive feedback on the revised manuscript and your recommendation for its publication. Your acknowledgment of the improvements in experimental detail, reconstructions, figures, and overall clarity is highly appreciated.

Thank you also for your constructive suggestion concerning the abstract. We have taken your advice to enhance clarity by revising the phrase to ”Here, we present a multi-peak Bragg coherent x-ray diffraction imaging (BCDI)” to accurately reflect the methodology used. This change aims to provide immediate clarity on our approach right from the abstract.

REVIEWERS' COMMENTS

Reviewer #1 (Remarks to the Author):

I appreciate the authors' thoughtful addressing to reviewers' comments. My concerns have been cleared in this revised version.